# Prenatal Diagnosis and Outcome of Tracheal Agenesis as Part of Congenital High Airway Obstruction Syndrome. Case Presentation and Literature Review

**DOI:** 10.3390/medicina57111253

**Published:** 2021-11-16

**Authors:** Tiberiu Georgescu, Viorica Radoi, Micaela Radulescu, Aurora Ilian, Oana Daniela Toader, Lucian G. Pop, Nicolae Bacalbasa

**Affiliations:** 1Department of Obstetrics and Gynecology, “Alessandrescu-Rusescu” National Institute of Mother and Child Care, 050474 Bucharest, Romania; tiberiuaugustin.georgescu@gmail.com (T.G.); viorica.radoi@yahoo.com (V.R.); radulescu_micaela@yahoo.com (M.R.); oana.toader@yahoo.com (O.D.T.); 2Department of Obstetrics and Gynecology, “Carol Davila” University of Medicine and Pharmacy, 050474 Bucharest, Romania; nicolaebacalbasa@gmail.com; 3Department of Obstetrics and Gynecology, “Victor Babes” University of Medicine and Pharmacy Timisoara, 300041 Timisora, Romania; aurora.ilian@umft.to

**Keywords:** tracheal atresia, hydrops, fetal demise

## Abstract

Tracheal atresia is an extremely rare condition whereby a partial or total obstruction of the trachea is seen. It is almost always lethal, with just a handful of cases that ended with a good outcome. In this study we report on a 15-week male fetus, diagnosed with hyperechogenic lungs, midline heart position and inverted diaphragm. Sonographic findings suggest congenital High Airway Obstruction Syndrome (CHAOS) An ultrasound scan and fetal MRI were not able to point out the exact obstruction level. In spite of extensive counselling, the parents opted to carry on with the pregnancy. Fetal demise was noted on a scan at 19 weeks gestation. After the elective termination of pregnancy, a post-mortem examination showed partial tracheal atresia with no other anomalies. Despite technological progress in CHAOS syndrome, a precise diagnosis and accurate prognosis remain elusive.

## 1. Introduction

Congenital high airway obstruction syndrome (CHAOS) is an extremely rare and usually lethal anomaly caused by congenital total or partial obstruction of the upper airways. First described by Hendrick et al., the acronym CHAOS is most suitable since a precise prenatal diagnosis, although desirable, is usually not possible. The true incidence of CHAOS is unknown, but the scientific literature reports about 1/50,000 [1]. Most of the lesions encompassed by this syndrome stem from the obstruction of the upper airways. Noteworthy is the sudden development of pulmonary hypertension, caused by the accumulation of lung fluids, followed by the compression and shifting in the heart’s position, which also features small dimensions. Under normal circumstances, the fluids secreted by the lung are eliminated in the amniotic cavity. Still, in patients with laryngeal or tracheal atresia, there is persistent retention of the liquid inside the lungs, accounting for the hyperechogenic ultrasound aspect and the gross hypervoluminous aspect of the lungs [2]. As a result, the heart is usually tiny, compressed and shifted.

Additionally, the intrathoracic pressure is increased, leading to cardiac failure, anaemia, ascites and hydrops. In this article, we report the case of a 20-week-old fetus with congenital high airway obstruction syndrome, which was diagnosed at 15 weeks, but the parents declined the termination of pregnancy as long as the fetus was alive. The postnatal autopsy revealed tracheal atresia.

## 2. Case Report

A 42-year-old patient with 2 previous live births (para 2) was referred to us for increased nuchal translucency of 3.2 mm following the first-trimester scan. Due to unforeseen reasons, she missed her appointment twice. When she finally presented to the doctor, the pregnancy was assessed at 15 weeks gestation, using convex probe 2D-4D RAB 6 D Voluson E10 BT16 (General Electric Healthcare, Ultrasound, Zipf, Austria), showing the following results: male sex fetus, with enlarged and echogenic lungs, flattened diaphragm, small and compressed heart, which was shifted toward the midline, large heart vessels and dilated bronchi (Figure 1, Figure 2 and Figure 3). There were minimum ascites and a regular quantity of amniotic fluid. Following counselling, the parents opted for a continuation of the pregnancy and amniocentesis with SNP microarray, which showed normal result (Agilent Cytogenomics 3.0.2.11 software). A further appointment was arranged at 17 weeks of gestation). The situation had worsened in the scan at 17 weeks, showing a hydropic baby, ascites, pericardial and pleural effusion). The parents rejected the option of pregnancy termination on religious grounds. The patient continued to be monitored weekly because the pregnancy had a high risk of fetal demise due to progressive hydrops. At 19 weeks of gestation, fetal death was noted on scan, as expected, and the parents finally agreed to proceed with the termination of the pregnancy. As she had 2 previous caesarean sections (10 years interpregnancy interval), prostaglandin doses were halved.

Permission for autopsy was obtained from the parents, and the product of conception with umbilical cord and placenta was sent to the Department of Pathology for histopathological examination.

The fetus that resulted from a therapeutic abortion was a male with the following anthropometric indices: weight: 200 g, cranio-caudal length: 12.5 cm, occipito-frontal circumference: 13 cm, thoracic circumference: 10 cm, abdominal circumference: 9 cm.

External examination revealed cyanotic skin and mucous membranes, normal represented subcutaneous tissue, non-palpable lymph nodes and normally shaped external genitalia. The attached umbilical cord measured 9.5/0.8 cm.

Anterior fontanelle measured 2.5/1 cm and posterior fontanelle measured 0.3/0.3 cm. Leptomeninges were semitransparent, with congestive vessels, revealing an edematous encephalon with symmetrical cerebral hemispheres and reddish spots on the cut surface which disappear when immersed underwater. Cerebellum measured 2/1/0.8 cm, with normally conformed and symmetrical cerebellar hemispheres. The bones at the base of the skull revealed no noticeable macroscopic changes.

The opening of the thoracic cavity revealed a grey-purple thymus, measuring 1.5/1/0.5 cm, with preserved lobulation. The esophagus was permeable, with congestive mucosa. The lungs were moderately enlarged, weighing 28 g each, with semitransparent pleura and ribs impressions on the external surface. Upon careful dissection of the upper respiratory system, airway obstruction was observed at the subglottic level, where dome-shaped cricoid cartilage appeared to be the cause of obstruction. No tracheoesophageal fistula was observed (Figure 4 and Figure 5). The heart measured 2.5/1.5/1 cm, with preserved veno-atrial, atrio-ventricular, and ventriculo-arterial correspondence. The diaphragm was flattened, with clear delimitation between the thoracic cavity and the abdominal cavity.

The opening of the abdominal cavity revealed approximately 5 mL of odorless serous liquid. The spleen, liver and kidney were enlarged and engorged with blood. Internal genitals were in accordance with external ones. No additional pathological findings were noted.

Histopathological evaluation showed tracheal atresia with overgrowth of the cricoid cartilage (Figure 6). Bilateral lungs showed marked dilatation of the alveolar spaces. No other associated anomalies were noted, and no genetic conditions were undertaken in this case. Based on the above findings, a diagnosis of CHAOS with distal tracheal atresia was confirmed.

## 3. Discussion

Chaos syndrome can be diagnosed on scan as early as 15 weeks as our case report showed. Laryngeal atresia is the most frequent cause but there are other etiologies such as tracheal atresia, laryngeal or tracheal webs, laryngeal cyst, subglottic stenosis or atresia, laryngeal or tracheal agenesis. Several cases where the fetus survived through fetoscopy, followed by caesarean section and EXIT (exutero intraprtum treatment) surgery, have been published [3]. EXIT surgery involves partial delivery of the baby which remains attached to the placenta, fetoplacental circulation being preserved. Intubation of fetus is attempted once. If it is unsuccessful tracheostomy below the level of obstruction is performed [4]. However, in the long run, numerous and redoubtable medical and surgical challenges remain. Overall, the prognosis is poor, especially in fetuses with associated anomalies and hydrops, where the outcome is invariable lethal. This condition is usually sporadic and in more than 90% of the cases, multiple congenital malformations are present. However, despite the progress made in genetic field, no casual gene has been found in tracheal agenesis patients so far. Tracheal agenesis might be isolated or a part of polymalformative syndromes, such as VACTERL (vertebral anomalies, anal atresia, cardiovascular anomalies, tracheoesophageal fistula, renal anomalies, and limb defects), TARCD syndrome (tracheal agenesis, radial and cardiovascular malformations, duodenal atresia), or Fraser syndrome, which is triggered by alterations in FRAS1 and FRAS2 genes [5,6]. Tracheal atresia has an incidence of 1/50,000 cases and has a worse prognosis than other causes of CHAOS syndrome. A literature review identified 17 cases of tracheal atresia/agenesis with known outcomes (Table 1).

The median gestational age of diagnosis was 21 weeks, and the median maternal age was 31 years (range 31–35). Our case was detected at 15 weeks of gestation, and according to our review this is the earliest gestational age at which CHAOS was diagnosed. All subjects showed typical signs of CHAOS. The median age at delivery was 29.5 weeks. Recent studies show that in selected cases, performing in utero-surgery with EXIT procedure at delivery can alter the usually grim prognosis of CHOAS. We found 5 cases of long-term surviving babies with tracheal atresia. Neurological development appears to be normal in fetuses without additional anomalies, but issues such as phonation or feeding can persist for the rest of their life. Hydrops is a sign of poor prognosis (also depicted in our case). Associated anomalies and particularly those defects that can’t be detected prenatally, complicate survival rate and prognosis. The complexity of the case can be enhanced by a twin pregnancy, where a normal twin is at risk of preterm delivery due to complications of the affected fetus (e.g., polyhydramnios). This matter is reported in articles published by Siemtka and by Ellio et all [10,13]. The case reported by Lim is one of the few cases in which hydrops were well tolerated by the fetus, allowing the delivery to take place at 32 weeks [17]. The fetus is a long-term survivor, being five years old at the time of publication. Cases of tracheal web can be managed by fetoscopy and are more prone to have a long-term survival, as it was showed by Saadi et al. [14]. A fetal MRI can help in making the right diagnosis, estimating the level of obstruction, and allowing a proper assessment of those patients that are suitable to be managed through in utero surgery and which are not. MRI investigation is also dependent to gestational age. In our case, the MRI performed at 17 weeks could not clarify the etiology of the CHAOS. But as other authors have published, it can be a handy tool for diagnosis, proper management, and counselling. The level of airway obstruction in tracheal atresia varies, and distal atresia with the upper half of the trachea and larynx being normal has been described before. Tracheal atresia is usually associated with tracheoesophageal or bronchoesophageal fistula, but isolated types have also been reported. It is possible that during pregnancy, prenatal features of CHAOS to resolve spontaneously as it was described by Roybal et al. [15]. This is possible when a fistula or a pinhole opening of the airway is present. As it can be seen from Table 1, there are cases that are thriving with comprehensive and long-term treatment. These are exceptions because, in general, tracheal atresia bears a grisly prognosis. Early hydrops, multiple anomalies and genes defects are worsening the outcome even more.

Most cases reported in the literature report a variety of histological alterations, although normal histologic aspects have also been reported. Most commonly described is the presence of congenital cystic adenomatoid malformation of the lung, especially in cases presenting with laryngeal atresia. A notable case of laryngeal atresia was described by Witters et al., who reported a case associated with pulmonary atresia, thymus hypoplasia and Fallot tetralogy, constellation consistent with the diagnosis of DiGeorge defect. Other issues presented with an increased number of alveoli, an aspect suggestive for the diagnosis of polyalveolar lungs while some authors report a subglotic stenosis, due to the hyperplasia of the fibrous connective tissue from the submucosa [2,7,18]. As we can see in Table 1, CHAOS is a second-trimester diagnosis. None of the cases described were picked up during the first-trimester scan. In a comprehensive study published by Syngelaki et al. l, an examination at 11–13 weeks scan was able to detect 27.6% of total malformations. Regarding thoracic malformations, first-trimester scan detected 29% cases of thoracic malformations but none of the congenital pulmonary airway malformation, congenital high-airway obstruction syndrome, mediastinal teratoma or pleural effusion. Due to the rarity of this condition, the study involved only one case of CHAOS, which was picked up during the second trimester. Although this is not a first trimester diagnosis, it is worth mentioning that our case, similar to the one published by Syngelaki et al.l, had the NT measurement above the 95% percentile [19]. This is an important clue, particularly when the paradigm of first-trimester screening is shifted toward cf DNA. In some countries, a first-trimester ultrasound scan is no longer part of the screening protocol [20,21,22,23]. Cf DNA screening without ultrasound will miss cases with increased NT, which is a valuable tool even in cases without apparent anomalies. Moreover, malformations that should always be detected such as acrania, alobar holoprosencephaly, exomphalos, gastroschisis and body-stalk anomaly will be overlooked, pushing and complicating their management at a later stage of gestation [24]. Therefore, we believe that cf DNA limits and benefits should be clearly explained to future parents as NIPT test provides, in some cases, a feeling of false reassurance. Emerging genomics methods have made possible the analysis of the whole genome by sequencing millions of cf-DNA fragments, both fetal and maternal. This makes diagnosis of monogenic possible through NIPT. Applying these technologies into clinical practice is a daunting task and the main goal in the genomic field. This brings not only opportunities but also many challenges. It is likely that many more patients with genetic disease will be identified and for this very reason, we must be prepared to offer appropriate, extensive, and empathic genetic counselling [25,26].

## 4. Conclusions

Our case showed an isolated anomaly of tracheal atresia diagnosed at 15 weeks of gestation. Parents option to carry on with the pregnancy allowed us to follow the natural course of the condition. However, the situation worsened at 19 weeks of gestation when fetal demise was noted on ultrasound examination.

Several years ago, CHAOS was considered universally lethal. Technological and medical progress have partially changed the fatal outcome in selected cases. The etiology of CHAOS is of utmost importance, primarily, since cases of tracheal pathology are known to have a worse prognosis than those of larynx atresia. The presence of a tracheal web or a natural fistula is an important factor in selecting patients for in utero intervention. EXIT procedure has proved helpful in improving the outcome of the pregnancy. Nevertheless, the prognosis of CHAOS syndrome is overall poor, and in many cases presence of multiple malformations or other genetic syndromes can be diagnosed only after birth. Therefore, a thorough discussion and counselling in a multidisciplinary format of the future parents are compulsory, requiring explanations regarding all options.

## Figures and Tables

**Figure 1 medicina-57-01253-f001:**
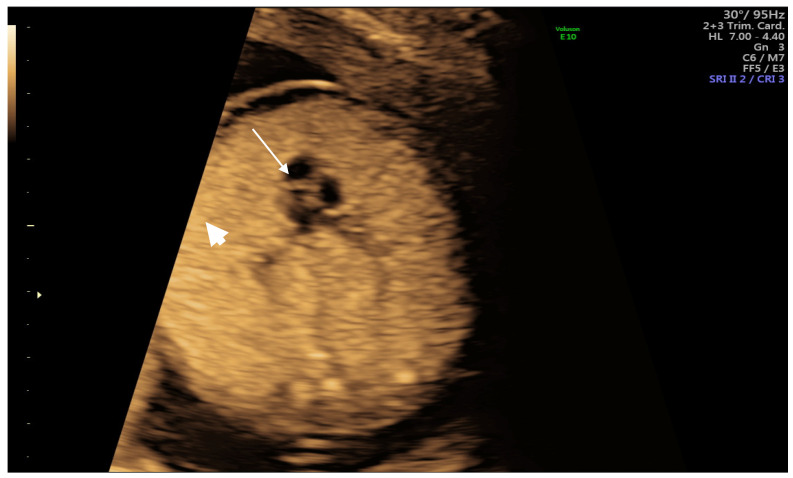
Heart (arrow) and hyperechogenic lungs (arrowhead) on scan.

**Figure 2 medicina-57-01253-f002:**
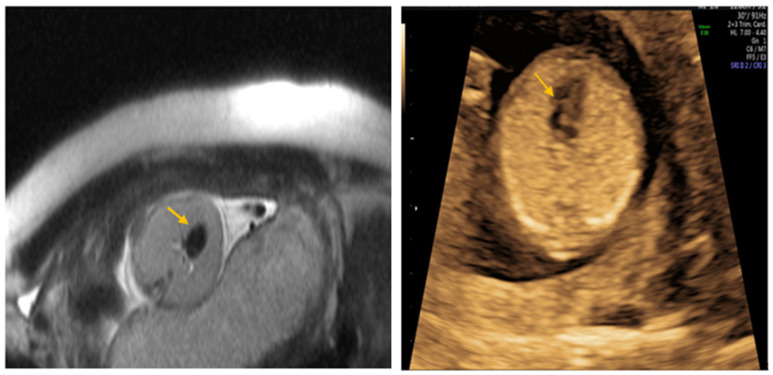
Compressed heart on MRI and ultrasound scan 16 weeks (arrow).

**Figure 3 medicina-57-01253-f003:**
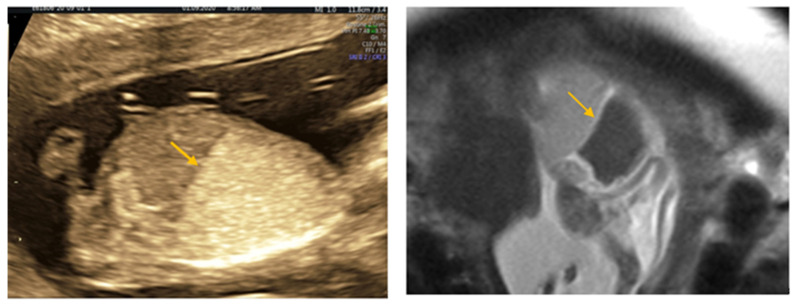
Inverted Diaphragm on ultrasound scan and on RMI (arrow).

**Figure 4 medicina-57-01253-f004:**
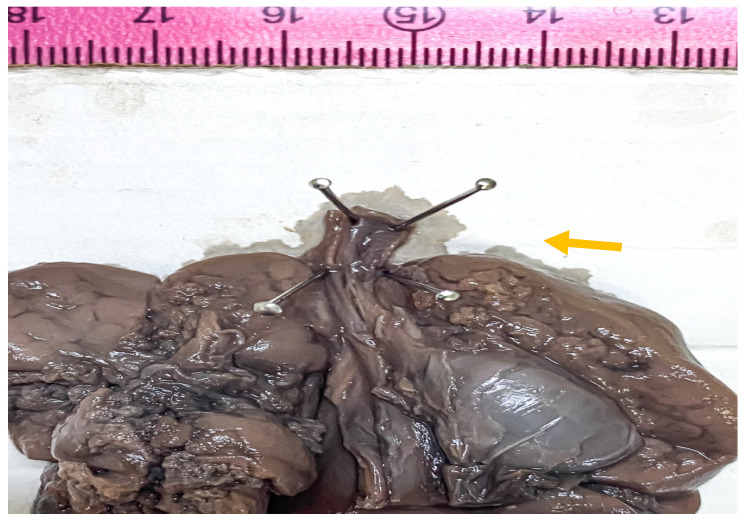
Gross aspect of the dissected cardiothoracic autopsy specimen in anterior view, showing opened distal trachea at the atretic level (black arrow). No bronchoesophageal fistula was detected.

**Figure 5 medicina-57-01253-f005:**
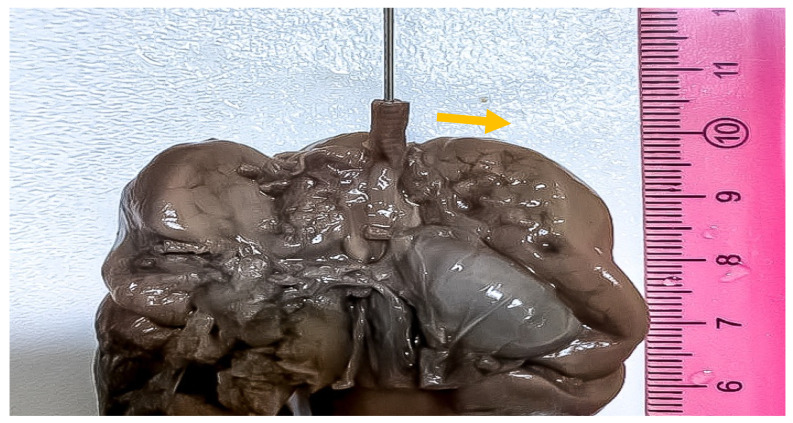
Gross aspect of the dissected cardiothoracic autopsy specimen in anterior view, showing imperforation level of the tracheal lumen, before opening (arrow).

**Figure 6 medicina-57-01253-f006:**
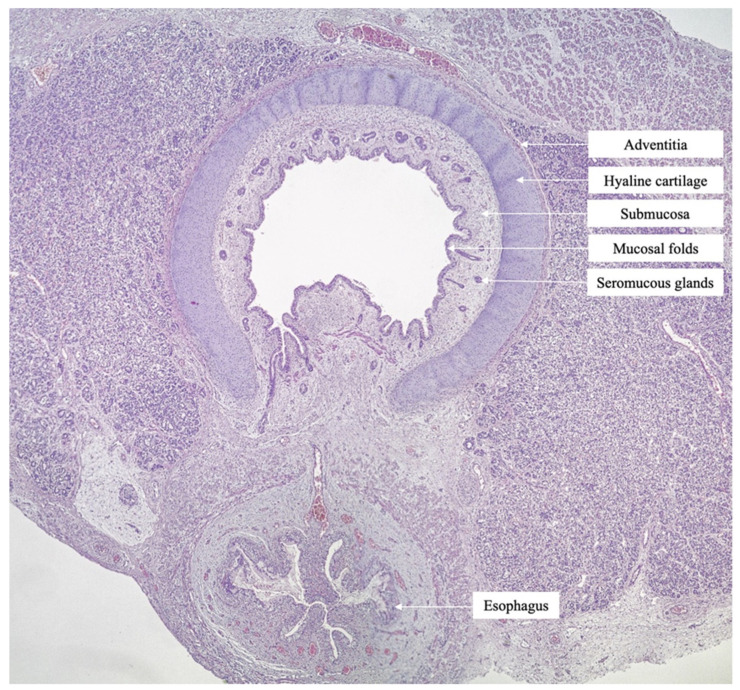
Histopathological aspect of the trachea with open lumen, above the obstructive level, composed of hyaline cartilage, tracheal mucosa, trachealis muscle and longitudinal muscle—behind the trachealis (H.E., 10×).

**Table 1 medicina-57-01253-t001:** Tracheal agenesis/atresia with known outcomes.

	Author	Year	Age	Weeks at Diagnosis	Weeks at Delivery	Outcome
1	Jong [7]	2020	31	23	37	waiting surgery
2	Umesh [8]	2017	27	24	24	termination
3	Kornacki [9]	2017	29	22	35	failed tracheostomy, died
4	Simetka [10]	2014	32	19 (DCDA)	31	exitus
5	Ulkumen [11]	2013	31	17	18	termination
6	Gonzales [12]	2018	30	24	33	laryngo tracheal reconstruction
7	Elliot [13]	2013	44	22 (DCDA)	36	laryngo tracheal reconstruction
8	Saadai [14]	2012	33	21	30	Fetoscopy, EXIT, thrieving
9			29	21	34	EXIT, ventilator dependant at 18 months
10				21	21	termination
11			21	21	35	prune belly sdr. ventilator dependant
12				19	19	termination
13	Roybay [15]	2010	N/A	20	31	tracheal reconstruction at 16 months
14		2010	N/A	20	31	failed exit, exitus
15	Oepkes [16]	2003	33	26	37	tracheoplasty, thrieving at 5 months
16	Lim [17]	2003	N/A	19	31	laryngo tracheal reconstruction, normal development at 5 years
17	present case	2021	40	15	19	exitus

EXIT—exutero intraprtum treatment; DCDA—Dicorionic diamniotic twins; N/A not available.

## Data Availability

Not applicable.

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
