# Peer review of "Prenatal Diagnosis and Outcome of Tracheal Agenesis as Part of Congenital High Airway Obstruction Syndrome. Case Presentation and Literature Review"

_medicina, 2021, doi:10.3390/medicina57111253_

Round 1

Reviewer 1 Report

This is a useful review of a rare fetal condition. The authors provide interesting personal images to illustrate the review.

Except from moderate English language changes / spell check (eg line 197 past tense worsened instead of present tense worsens).

Author Response

Answer to reviewer 1

Dear Sir / Madam

Thank you very much for reviewing our article and your observations that are improving the quality of the paper considerably.

  1. I will answer your observation as follows- Spelling has been checked and corrected

Reviewer 2 Report

In this paper the authors reported the case of a 20-week-old fetus with 
congenital high airway obstruction syndrome (CHAOS syndrome), which was diagnosed at 15 weeks, but the parents declined the termination of pregnancy as long as the fetus was alive. The postnatal autopsy revealed tracheal atresia. CHAOS is mostly sporadic, and the exact incidence is not known. Although the described case could allow to improve the prenatal definition of CHAOS syndrome with the hope of neonatal outcome improvements, I suggest some  revisions.

Major revisions: 

  • The authors refer to table 1 in the text, but there is no table reported.
  • In the discussion the authors reported all the cases described in the literature, but they missed to give their own opinion.
  • Because the authors reported literature data, I suggest to add possible genetic causes of CHAOS syndrome

Minor :

  • Figure 1 and no Fig 1
  • In the figure 1 the arrow was not on the immagine
  • lane 31: add a space between 1/50000 and (1)
  • lane 125 replace fig4 with figure 4
  • lane 181: add a point before it

Author Response

Answer to reviewer 2:

Dear sir,

Thank you very much for reviewing our article and your observations that are improving the quality of the paper considerably.

I will answer to your observation as follows:

     1. Table 1 is on page 7. Its absence was noticed by another reviewer as well, although it was sent together with the rest of the manuscript.

  1. A paragraph on genetic involvement in CHAOS has been added (line 150-to 157)
  2. Figure 1 spelling was corrected, and arrow rearranged
  3. space has been added

      5. Figure 4 instead of fig 4

      6. point has been added 

Reviewer 3 Report

please find the attached review report 

Author Response

Answer to reviewer 3:

Dear Madam or Sir,

Thank you for reviewing this manuscript and for sharing your thoughts in connection with the paper. We have made corrections and revised the text in line with your suggestions and we believe this has assisted in substantially improving the clarity of the text and the overall quality of the paper.

Enclosed, please find responses to your individual points and the revised manuscript

  1. We concur with the opinion that first-trimester ultrasound scan is of paramount importance in prenatal care and in an era when technological advances are faster than clinical advances and there is a validation time lag. A paragraph regarding the importance of first-trimester ultrasound has been added, followed by  a discussion regarding limits of NIPT and  future of NIPT ( monogenic conditions)  pg 8 ( yellow colour) 

  1. Table 1 is at page 7. Its absence was noticed by another reviewer as well, although it was sent together with the rest of the manuscript

  1. NT was above the 95 th centile. According to our review, this was the first case diagnosed at such an early gestational age, which is now mentioned in the text at line 163. Thank you for your tip.

  1. Exit procedure has been briefly described in lines 143 to 146

  1. “Et al “has been added after each citation

  1. Typos and grammatical inconsistencies have been reviewed